# Better programmatic outcome with the shorter regimen for the treatment of multidrug-resistant tuberculosis (MDR-TB) in Guinea: A retrospective cohort study

Souleymane Hassane-Harouna[1*‡], Gba-Foromo Cherif[1‡], Nimer Ortuno-Gutierrez[2], Diao Cisse[1], Lansana Mady Camara[3,4], Boubacar Djelo Diallo[4], Souleymane Camara[5], Adama Marie Bangoura[5], Lutgarde Lynen[6], Tom Decroo[6,7]

1 Action Damien, Conakry, Guinea, 2 Action Damien, Research Department in Brussels, Brussels, Belgium, 3 Gamal Abdel Nasser University, Conakry, Guinea, 4 Pneumo Phtisiologie, Hôpital Universitaire Ignace Deen, Conakry, Guinea, 5 National Tuberculosis Control Program, Conakry, Guinea, 6 Institute of Tropical Medicine, Antwerp, Belgium, 7 Research Foundation Flanders, Brussels, Belgium

‡ These authors are joint first authors.
* hassoul20@yahoo.fr

**Data Availability Statement:** The data supporting the findings of this publication are retained at the

## Abstract

### Setting

Since August 2016, after the Ebola outbreak, the Guinean National Tuberculosis Programme and Damien Foundation implemented the shorter treatment regimen (STR) for multidrug-resistant tuberculosis (MDR-TB) in the three MDR-TB sites of Conakry. Previously, the longer regimen was used to treat MDR-TB.

### Objectives

In a post-Ebola context, with a weakened health system, we describe the MDR-TB treatment uptake, patients characteristics, treatment outcomes and estimate the effect of using the longer versus STR on having a programmatically adverse outcome.

### Design

This is a retrospective cohort study in RR-TB patients treated with either the longer regimen or STR.

### Results

In Conakry, in 2016 and 2017, 131 and 219 patients were diagnosed with rifampicin-resistant tuberculosis (RR-TB); and 108 and 163 started treatment, respectively. Of 271 patients who started treatment, 75 were treated with the longer regimen and 196 with the STR. Patients characteristics were similar regardless of the regimen except that the median age was higher among those treated with a longer regimen (30 years (IQR:24–38) versus 26 years (IQR:21–39) for the STR. Patients treated with a STR were more likely to obtain a programmatically favorable outcome (74.0% vs 58.7%, p = 0.01) as lost to follow up was

Institute of Tropical Medicine, Antwerp and will not be made openly accessible due to ethical and privacy concerns. Data access requests can be sent to the Institute of Tropical Medicine at ITMresearchdataaccess@itg.be.

**Funding:** The authors received no specific funding for this work.

**Competing interests:** The authors have declared that no competing interests exist.

higher among those treated with a longer regimen (20.0% vs 8.2%, p = 0.006). Patients on a longer regimen were more than 2 times more likely (aOR: 2.5; 95%CI:1.3,4.7) to have a programmatically adverse outcome as well as being 45 years or older (aOR: 2.8; 95% CI:1.3,6.2), HIV positive (aOR:3.3; 95%CI:1.6,6.6) and attendance at a clinic without NGO support (aOR:3.0; 95%:1.6,5.7).

## Conclusion

In Guinea, patients treated with the STR were more likely to have a successful outcome than those treated with the longer MDR-TB treatment regimen. Lost to follow-up was higher in patients on the longer regimen. However, STR treatment outcomes were less good than those reported in the region.

## Introduction

Diagnosis and treatment of rifampicin-resistant tuberculosis (RR-TB) remain major public health challenges. Globally in 2018, 484 000 people developed TB that was resistant to rifampicin (the most effective first-line drug) and of these, 78% had multidrug-resistant TB (MDR-TB). The treatment success rate still remains poor (56%) [1]. In 2016, World Health Organization (WHO) recommended a shorter regimen (STR) lasting 9 to 12 months for RR/MDR-TB treatment [2].

In Guinea, according WHO, there are an estimate 680 incident cases of RR-TB. Diagnosis of RR-TB relies essentially on Xpert MTB/RIF testing in previously treated TB cases and contacts of confirmed RR-TB index cases [3]. In 2016, when Guinea was recovering from the devastating Ebola Virus Disease (EVD) outbreak, the introduction of the STR was expected to reduce pressure on both patients and the health system. The STR was reported to result in over 80% programmatic success in nine West and Central African countries using normal-dose moxifloxacin [4], even though the Bangladesh STR relied on high-dose gatifloxacin as core drug (the drug driving the efficacy of the regimen) [5]. In Guinea, as gatifloxacin was not available anymore [6], the STR was built on high-dose moxifloxacin instead of high-dose gatifloxacin as core drug.

No previous study has evaluated the use of a high-dose moxifloxacin STR in programmatic conditions. In the present study, we will describe: 1) the number of patients diagnosed with RR-TB in Guinea and treated between 2016 and 2017. 2) the treatment outcomes and estimate of the effect of using either the longer or shorter treatment regimen.

## Methods

### Design

This is a retrospective cohort study.

### Setting

Guinea is a West African country with an estimated population of 12 million. WHO estimates that every year 680 new patients develop resistance to rifampicin TB [1].

This study used data from all three sites involved in RR-TB care in the country's capital, Conakry, between 2016 and 2017: Ignace Deen hospital, the TB referral health center Carrière

and Tombolia Health Center. Since its implementation in 2011, Xpert MTB/RIF testing has considerably increased to date due to the availability of more GeneXpert machines in the country. The Tombolia Health Center receives support from the Damien Foundation, a Belgian non-governmental organization (NGO) experienced in TB and leprosy's care. This support includes a monthly nutritional package, transport to the health facility fees, and clinical and biological examination free of charge for all MRD/RR-TB patients.

## Study population

All patients diagnosed with rifampicin-resistant pulmonary tuberculosis and treated between 2016 and 2017 in Guinea were included in the study.

## Treatment regimens

Between January and July 2016, only the longer regimen was used. Since August 2016 the STR replaced the longer regimen, while patients already enrolled on the longer regimen continued the same longer regimen. Implemented in 2008, the longer regimen (18–24 months) consisted of 6 months of kanamycin (Km), levofloxacin (Lfx), cycloserin (Cs), prothionamide (Pto) and pyrazinamide (Z), followed by 18 months of Lfx, Pto, Cs and Z (6Km-Lfx-Cs-Pto-Z/18 Lfx-Pto-Cs-Z). The STR regimen, included 4 months of Km, high-dose moxifloxacin (Mfxh), Pto, high-dose isoniazide (Hh), clofazimine (Cfz), ethambutol (E) and Z, followed by 5 months of Mfxh, Cfz, E, and Z (4–6 Km-Mfxh-Pto-Hhd-Cfz-E-Z /5 Mfxh-Cfz-E-Z). The intensive phase can be extended to 6 months when the sputum conversion is not obtained at the end of the 4th or 5th month of treatment [7].

Ambulatory care was delivered for patients with a good clinical condition while severely ill patients were hospitalized in the Ignace Deen hospital.

Active drug safety monitoring and management (aDSM) was done but data was not systematically collected in patients treated with the longer regimen.

## Data variables, sources and definitions

Variables included age, sex, HIV status, MDR-TB clinic, date of treatment start, treatment regimen, and treatment outcomes.

WHO treatment outcomes (cure, treatment completion, death, treatment failure, lost to follow-up and not evaluated) were used [8]. Programmatically adverse outcomes were death, treatment failure and lost to follow-up. Programmatically favorable outcomes included cure and treatment completion.

## Data collection and analysis

Study data were retrieved from the national Excel® database used to routinely monitor RR-TB treatment and from individual patient files. Data were routinely entered into this database and monthly updated according to patient files. A quality check was performed by National TB Programme and Damien Foundation staff. Patient demographics, clinical characteristics and treatment outcomes were described using counts and proportions.

We calculated medians and interquartile ranges for continuous variables. We used the chi-squared test and the Wilcoxon rank-sum test to compare categorical variables and continuous variables. Univariate and multivariate logistic regression were used to estimate crude and adjusted odds ratios for having a programmatically adverse outcome. The saturated multivariate model was simplified, until all variables in the final model were significant at level 0.05. We used Stata (version 14.2) for analysis.

## Ethics

The Guinea National Tuberculosis Programme approved conducting the present retrospective study of routinely collected data. Co-authors include National Tuberculosis Programme staff. Informed consent was obtained from all participants before starting treatment. The Institutional Review Board of the Institute of Tropical Medicine Antwerp approved the study protocol. Confidentiality was assured throughout.

## Results

In Guinea, in 2016, 131 patients were diagnosed with RR-TB and 108 started treatment. In 2017, 219 patients were diagnosed with RR-TB and 163 started treatment. All 271 patients started on MDR-TB treatment were included in the analysis. Of 271, 75 were started on a longer regimen, and 196 on the STR.

The proportion of patients that were male, HIV positive, and attending a clinic supported by an NGO was similar regardless of the regimen provided. Median age was higher among those treated with a longer regimen (30 years;IQR:24–38) compared to those treated with the shorter regimen (26 years; IQR:21–39) (Table 1).

Table 3 shows that patients on a longer regimen were more than 2 times more likely (aOR: 2.5; 95%CI:1.3,4.7) to have a programmatically adverse outcome. Other factors associated with having a programmatically adverse outcome were: being 45 years or older (aOR: 2.8; 95% CI:1.3,6.2), being HIV positive (aOR:3.3; 95%CI:1.6,6.6), and attendance at a clinic without NGO support (aOR:3.0; 95%:1.6,5.7).

**Table 1. Characteristics among patients who started MDR-TB treatment between 2016 and 2017.**

|  | | Treatment regimens | | | |
|---|---|---|---|---|---|
|  | **Total** | **Longer** | **%** | **Shorter n = 196** | **%** | **p-value** [a] |
|  | **n = 271** | **n = 75** | | | | |
| **Sex** | | | | | | 0.9 |
| Female | 76 | 21 | 28.0 | 55 | 28.1 | |
| Male | 195 | 54 | 72.0 | 141 | 71.9 | |
| **Age group (years) (n = 270)** | | | | | | 0.1 |
| 0–14 | 4 | 2 | 2.7 | 2 | 1.0 | |
| 15–29 | 138 | 46 | 61.3 | 92 | 46.9 | |
| 30–44 | 87 | 19 | 25.3 | 68 | 34.7 | |
| >45 | 42 | 8 | 10.7 | 34 | 17.3 | |
| Age, median (IQR) | 28 (23–38) | 30 | (24–38) | 26 | (21–39) | 0.02 |
| **HIV status (n = 270)** | | | | | | 0.9 |
| Negative | 222 | 63 | 84.0 | 159 | 81.1 | |
| Positive | 49 | 12 | 16.0 | 37 | 18.9 | |
| **NGO support to the clinic where care is provided** | | | | | | 0.7 |
| Yes | 102 | 27 | 36.0 | 75 | 38.3 | |
| No | 169 | 48 | 64.0 | 121 | 61.7 | |

IQR: interquartile range, n = number.

[a] Chi-squared test for categorical variables, Wilcoxon rank-sum test for age.

Table 2 shows that patients treated with a shorter regimen were more likely to obtain a programmatically favorable outcome (74.0% vs 58.7%, risk difference 15.3% (95% CI:2.6–28.0), p = 0.01). Lost to follow up (LTFU) was higher among those treated with a longer regimen (20.0% vs 8.2%, risk difference 11.8% (95%CI 2.0–21.7), p = 0.006).

**Table 2. Treatment outcomes among 271 patients started on MDR-TB treatment in Guinea, 2016 and 2017.**

| | Treatment regimens | |
|---|---|---|
| | **Longer** | **Shorter** |
| | **n = 75** | **n = 196** |
| Cured, n(%) | 42 (56) | 112 (57.1) |
| Treatment completed, n(%) | 2 (2.7) | 33 (16.8) |
| Died, n(%) | 11(14.7) | 30(15.3) |
| Treatment failure, n(%) | 5(6.7) | 5(2.6) |
| Lost to follow-up, n(%) | 15(20.0) | 16(8.2) |
| Programmatically favorable$ | 44(58.7) | 145(74.0) |
| Programmatically adverse | 31(41.3) | 51(26.0) |

$ Programmatically adverse outcomes included died, treatment failure, and lost to follow-up, favorable outcomes included cured and treatment completed

## Discussion

This is the first study comparing a high-dose moxifloxacin STR with a longer MDR-TB treatment regimen in a programmatic setting. Our findings complement those of the STREAM trial, which also compared a high-dose moxifloxacin STR with a longer MDR-TB treatment

**Table 3. Predictors of an programmatically adverse treatment outcome, among 271 patients started on MDR-TB treatment in Guinea, between 2016 and 2017.**

| | Total | N | (%) | OR | [95%CI] | aOR | [95%CI] |
|---|---|---|---|---|---|---|---|
| | | | | **Unfavorable outcome$** | | | |
| **Total** | **271** | **82** | **30.3** | **NA** | | **NA** | |
| Sex | | | | | | NS | |
| Female | 76 | 26 | 34.2 | 1 | | | |
| Male | 195 | 56 | 28.7 | 0.8 | [0.44,1.37] | | |
| Age groups | | | | | | | |
| <15 | 4 | 3 | 75.0 | 1 | | 1 | |
| 15–30 | 138 | 31 | 22.5 | 10.4* | [1.04,103.10] | 5.5 | [0.4,67.0] |
| 30–44 | 87 | 30 | 34.5 | 1.8* | [1.00,3.30] | 1.6 | [0.9,3.2] |
| 45-. . . | 42 | 18 | 42.9 | 2.6* | [1.25,5.37] | 2.8** | [1.3,6.2] |
| HIV status | | | | | | | |
| Negative | 222 | 57 | 25.7 | 1 | | 1 | |
| Positive | 49 | 25 | 51 | 3.0*** | [1.60,5.70] | 3.3*** | [1.6,6.6] |
| NGO support to clinic where care is provided | | | | | | | |
| Yes | 102 | 18 | 17.6 | 1 | | 1 | |
| No | 169 | 64 | 37.9 | 2.8*** | [1.57,5.16] | 3.0*** | [1.6,5.7] |
| Regimen | | | | | | | |
| Shorter | 196 | 51 | 26.0 | 1 | | 1 | |
| Longer | 75 | 31 | 41.3 | 2.0* | [1.14,3.51] | 2.5** | [1.3,4.7] |

* p < 0.05

** p < 0.01

*** p < 0.001.

NA: not applicable; NS: not significant.

$Programmatically adverse outcomes included died, treatment failure, and lost to follow-up, favorable outcomes included cured and treatment completed.

regimen [9]. The STREAM trial showed 79.8% success among those treated with a longer regimen versus 78.8% success among those treated with the STR, and concluded that the STR was non-inferior compared to the longer regimen (1.0% difference, with the 95% CI not exceeding the non-inferiority margin) [9]. Our study showed 58.7% vs 74.0% success, with a significant difference of 15.3% (95%CI:2.6–28.0), in favor of the shorter regimen. This difference between both cohorts was mainly explained by a higher proportion of patients reported as lost to follow-up (LTFU) among those treated with a longer regimen (20.0% vs 8.2%) [9, 10]. Our findings are coherent with those from a recent review, which also showed that patients treated with the STR were less likely to be LTFU than those treated with the longer MDR-TB treatment regimen [11].

Our findings are important as they facilitate the interpretation of the STREAM results. The STREAM trial, as most trials, was conducted in circumstances that are better controlled than in routine practice. Among STREAM trial patients started on the longer regimen and included in the efficacy analysis, 2.4% (3/124) were LTFU versus 0.4% (1/245) among those treated with the STR [9, 10]. This contrasts with global 2019 WHO update on MDR-TB, showing the reality of National Tuberculosis Programmes, and reporting 21% LTFU among those started on the longer regimen [12]. The STREAM findings also contrast with our study findings, showing that 20.0% of those on a longer regimen were LTFU. Of note, the proportion LTFU in our study was very similar to the 21% reported globally [12]. We speculate that the STREAM trial setting very likely affected treatment outcomes: patients who would have been LTFU in the real world were disproportionally retained in care in clinical trial conditions, in favor of the longer regimen [9, 10].

WHO 2018 data also show that 15% of patients treated with a longer regimen died, 8% were reported with treatment failure, and only 55% were treated successfully [12]. Similarly, in Guinea, of patients treated with a longer regimen 20.0% were LTFU, 14.7% died, and 6.7% had treatment failure, and only 58.7% were treated successfully. We can conclude that treatment outcomes of patients treated with a longer regimen in our setting are very similar to these global figures, reported for the same period.

However, treatment outcomes among patients treated with the shorter regimen were less good than those reported elsewhere in the Western African region [4]. In our study 74% were treated successfully, 15.3% died, 8.2% were LTFU, and 2.6% experienced treatment failure. In a study summarizing data from 1006 patients from nine West-African countries (which did not include Guinea) treated with a normal-dose moxifloxacin STR, 81.6% were treated successfully, 7.8% died, 4.8% were LTFU and 5.9% had treatment failure [4].

How to explain the lower proportion of programmatic success achieved in Guinea? We speculate that the higher level of mortality and LTFU may be caused by multiple factors, including delayed diagnosis or delayed treatment initiation, lower level of patient support, and/or lower level of quality of clinical care. Even if patients in the study were enrolled on treatment between 2016 and 2017 in the "post-Ebola" period, as a previous showed that the outbreak had little effect on TB programme performance [13].

Another predictor of adverse outcomes is initial resistance to fluoroquinolone [5] Programmatic success in patients treated with a normal-dose moxifloxacin STR in Swaziland and Uzbekistan was 71% and 70%, respectively. Bacteriologically adverse outcomes, either failure or relapse, were as frequent as 15% and 16.7% [14]. Initial resistance to drugs used in the STR, especially the fluoroquinolone, may explain these poor results [5, 14]. Even though initial resistance to fluoroquinolone was not tested systematically, we speculate that initial resistance does not explain the outcomes shown in Guinea, as only 2.6% of patients were reported with treatment failure, much lower than in the above mentioned settings with a high prevalence of fluoroquinolone resistance.

The proportion of patients with treatment failure was lower in the Guinea cohort than in the cohort showing data from nine African countries, where normal-dose moxifloxacin was used. The effect of moxifloxacin on microbiological kill is known to be dose-dependent [15]. But more important than the dosing of moxifloxacin, recent evidence shows that the choice of fluoroquinolone affects treatment outcomes [16]. In cohorts using a gatifloxacin-based STR, programmatic success ranged between 84.5% and 89%, with 1.4% (7/515) and 0.7% (1/150) treatment failure in Bangladesh and in Cameroon, respectively [5, 17, 18]. Gatifloxacin-based STR perform better than moxifloxacin-based STR in terms of bacteriological outcomes [16]. A recent study showed that a moxifloxacin-based STR had a 8.4-fold times larger odds of having an adverse bacteriological outcome. Moreover, none of the patients on a gatifloxacin-based regimen developed resistance to fluoroquinolone, while 4 of 228 patients with initially fluoro-quinolone- susceptible TB developed resistance [16].

Our study has some important limitations. Data on conditions that could affect outcomes, such as alcoholism, diabetes and chronic obstructive pulmonary disease, were not systemati-cally collected, and therefore not included in the analysis. Phenotypic or genotypic drug sus-ceptibility testing was not available to exhaustively determine the initial resistance profile; hence the effect of initial resistance on treatment response was not assessed. We did also not present data on the safety of the longer or shorter regimen, as these were not collected system-atically. However, the STREAM trial studied the same regimens and showed that both regi-mens resulted in a similar frequency of adverse events [9]. Another limitation is the enrollment of all patients on the longer regimen in 2016 while patients treated with the STR were enrolled in both 2016 and 2017. On the other hand, given the long duration of the longer regimen, the treatment period of both cohorts overlapped as patients on the longer regimen were treated until 2018, making systematic bias less likely. Moreover, the MDR-TB pro-gramme was stable between 2016 and 2018, relying on Xpert MTB/RIF for RR-TB diagnosis, and the treatment regimens described in this manuscript. Finally, inherent to the design of the study, patients enrolled in the longer and shorter regimen may not have been entirely compa-rable. On the other hand, measured baseline characteristics did not identify relevant differ-ences between both cohorts. Another strength of the study is its exhaustive national sample and the use of routinely collected data. Hence, our findings are generalizable for the Guinean setting.

## Conclusion

In Guinea, programmatic success was higher when patients were treated with a high-dose moxifloxacin-based STR, when compared with a longer MDR-TB treatment regimen. How-ever, outcomes of the STR cohort were less good than those reported elsewhere in the Western African region. Our findings, showing the reality of the National Tuberculosis Programme in Guinea, complement findings of the STREAM trial, the only other study that evaluated a high-dose moxifloxacin-based STR against a longer regimen.

## Acknowledgments

The authors would like to acknowledge the staff members of National Reference Laboratory for Mycobacteriology of Conakry. The authors would also like to thank all physicians, nurses and Damien Foundation staff members involved in multidrug-resistant tuberculosis care as well as the patients for their cooperation and participation in this study.

Finally, let us express all our gratitude to the Belgian Directorate-General for Development (DGD) scholarship program.

## Author Contributions

**Conceptualization:** Souleymane Hassane-Harouna, Gba-Foromo Cherif, Lansana Mady Camara, Tom Decroo.

**Data curation:** Souleymane Hassane-Harouna, Gba-Foromo Cherif, Diao Cisse, Souleymane Camara, Tom Decroo.

**Formal analysis:** Souleymane Hassane-Harouna, Gba-Foromo Cherif, Nimer Ortuno-Gutierrez, Lutgarde Lynen, Tom Decroo.

**Investigation:** Souleymane Hassane-Harouna, Gba-Foromo Cherif, Diao Cisse.

**Methodology:** Souleymane Hassane-Harouna, Gba-Foromo Cherif, Nimer Ortuno-Gutierrez, Lansana Mady Camara, Adama Marie Bangoura, Lutgarde Lynen, Tom Decroo.

**Software:** Tom Decroo.

**Supervision:** Souleymane Hassane-Harouna, Gba-Foromo Cherif, Nimer Ortuno-Gutierrez, Lansana Mady Camara, Boubacar Djelo Diallo, Adama Marie Bangoura, Lutgarde Lynen, Tom Decroo.

**Validation:** Souleymane Hassane-Harouna, Gba-Foromo Cherif, Souleymane Camara, Tom Decroo.

**Visualization:** Souleymane Hassane-Harouna, Gba-Foromo Cherif, Tom Decroo.

**Writing – original draft:** Souleymane Hassane-Harouna, Gba-Foromo Cherif, Nimer Ortuno-Gutierrez, Diao Cisse, Boubacar Djelo Diallo, Lutgarde Lynen, Tom Decroo.

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
