## [Decision Letter · Decision Letter 0]

29 Jun 2020

PONE-D-20-12232

Better programmatic outcome with the shorter regimen for the treatment of multidrug-resistant tuberculosis (MDR-TB) in Guinea: a retrospective cohort study

PLOS ONE

Dear Dr. Harouna,

Thank you for submitting your manuscript to PLOS ONE. After careful consideration, we feel that it has merit but does not fully meet PLOS ONE’s publication criteria as it currently stands. Therefore, we invite you to submit a revised version of the manuscript that addresses the points raised during the review process.

We look forward to receiving your revised manuscript.

Kind regards,

Hasnain Seyed Ehtesham

Academic Editor

PLOS ONE

Journal Requirements:

Additional Editor Comments (if provided):

Major Revision

Reviewers' comments:

Reviewer's Responses to Questions

**Comments to the Author**

1. Is the manuscript technically sound, and do the data support the conclusions?

Reviewer #1: Yes

Reviewer #2: Yes

2. Has the statistical analysis been performed appropriately and rigorously? 

Reviewer #1: Yes

Reviewer #2: Yes

3. Have the authors made all data underlying the findings in their manuscript fully available?

Reviewer #1: Yes

Reviewer #2: Yes

4. Is the manuscript presented in an intelligible fashion and written in standard English?

Reviewer #1: Yes

Reviewer #2: Yes

5. Review Comments to the Author

Reviewer #1: Souleymane Hassane Harouna et al analyzed the response of RR TB patients for various treatment regimen and compared the efficacy of short tern regimen with long term treatment outcome .

This is sound clinical study which address various issues related to the resistance / sensitivity of the patients toward anti TB drug regimens.

I have following concern which is related to the microbiological aspect of the study and feel that addressing this would enhanced the quality of the study multi-folds

1. The author should clarify the exclusion criteria for the patients which they have selected , How many patients had Aspergillosis, COPD and diabetes because these traits can influence the outcome of the treatment and this is believed to be responsible for the adverse outcome during long term treatment regimen.

2. I would advice author to select 5 patients in both STR and LTR and analyze whole genome analysis of bacterial genes because i feel that mycobacterial devR and dosR responsive element may also provide a microbiological correlation with respect to the response of the patients

3. Considering the resistance of patients for LTR , whether inclusion of Rapamycin or bedaquiline can change the response in LTR group.

4. Immunological nitch is linked to latency / dormancy related phenotype in TB so from that analyzing few patients with foamy macrophage, MDSC , Treg along with HLA-G pattern may add to the immune mediated mechanism responsible for adverse response in LTR groups

Reviewer #2: Comments:

Harouna et al. written about the treatment outcome for shorter treatment regimen and longer regimen categories with modified treatment in existing treatment regimen. Data proved the treatment utility and recommendations in present scenario for TB programme successfully progresses worldwide. Following are some suggestions which can make the manuscript stronger and beneficial for the readers:

Abstract:

• Abbreviation of rifampicin resistance (RR) should come upon first appearance in the text.

Introduction:

• Introduction should modify because author had discussed previous trail related to the present study. That would be a part of discussion only. Line no. 60 to 72

• Reference No. 1 should be updated with recent one.

• It’s little bit inequality and confusing regarding the sentences for the dose of MFX used in the study. Line no. 57 (This study used normal-dose moxifloxacin.) indicated normal dose while, in Line no. 60 (In Guinea, the STR relied on high-dose moxifloxacin instead) high dose MFX is mentioned.

• In first objective, author had mentioned about the diagnosis and number of RR-TB. In that view, author should write few points about the prevalence and diagnostics modalities used in his country.

Method: It observed that author/s ware collected the data very precisely. However, required some information to improve manuscript like:

• Delete “longer” (line no. 99, continued the same longer regimen.).

• Add references regarding the doses and drugs for STR and longer regimen. (Line no. 97-106).

• Author should point out the type (pulmonary or extra-pulmonary) of tuberculosis among patient requited for this study.

• In patient history, alcoholism and smoker status of patient as well as house hold contact should be noted if possible. Because these habits are strongly associated with treatments which leads to recurrence.

Discussion:

• Provide reference for these sentences. Like:

“The STREAM trial showed 79.8% success among those treated with a….”

“We speculate that the STREAM trial setting very likely affected treatment..”

“However, treatment outcomes among patients treated with the shorter regimen ..”

“But more important than the dosing of moxifloxacin, recent evidence shows that the choice of fluoroquinolone..”

• Author should modify the sentence “Another predictor of treatment success is initial resistance to fluoroquinolone”, “Bacteriologically adverse outcomes, either failure or relapse, were as frequent as 15% and 16.7%”.

• Authors are advised to cite some recent studies found relevant to the present study.

Like:

Conclusion: Authors are advised to reframe the sentence (Although better than outcomes from patients…..)

6. PLOS authors have the option to publish the peer review history of their article (what does this mean?). If published, this will include your full peer review and any attached files.

Reviewer #1: No

Reviewer #2: No

---

## [Author Response · Author response to Decision Letter 0]

15 Jul 2020

Reviewer #1: 

Souleymane Hassane Harouna et al analyzed the response of RR TB patients for various treatment regimen and compared the efficacy of short tern regimen with long term treatment outcome .

This is sound clinical study which address various issues related to the resistance / sensitivity of the patients toward anti TB drug regimens.

I have following concern which is related to the microbiological aspect of the study and feel that addressing this would enhanced the quality of the study multi-folds

RESPONSE: Many thanks for the constructive feedback. 

1. The author should clarify the exclusion criteria for the patients which they have selected , How many patients had Aspergillosis, COPD and diabetes because these traits can influence the outcome of the treatment and this is believed to be responsible for the adverse outcome during long term treatment regimen.

RESPONSE: This study describes the routine management of multidrug resistant tuberculosis in Guinea. Thus all patients treated in the study period were included as said in the method section. None of these patients had Aspergillosis. 

Indeed, COPD and diabetes may affect the outcome. Unfortunately, these data were not systematically collected for all patients, hence not included in the analysis. We therefore expanded the limitations section in the discussion as follows: “Data on conditions that could affect the outcomes, such as alcoholism, diabetes and chronic obstructive pulmonary disease, were not systematically collected, and therefore not included in the analysis.”(line 238)

2. I would advice author to select 5 patients in both STR and LTR and analyze whole genome analysis of bacterial genes because i feel that mycobacterial devR and dosR responsive element may also provide a microbiological correlation with respect to the response of the patients

RESPONSE: This retrospective study analyses data collected in routine practice. As our lab is not able to perform whole genome sequencing (WGS), WGS data (or other tests to exhaustively determine the resistance profile) were not available. This point also will be added as a limitation: “Phenotypic or genotypic drug susceptibility testing was not available to exhaustively determine the initial resistance profile; hence the effect of initial resistance on treatment response was not assessed” Line 240

3. Considering the resistance of patients for LTR , whether inclusion of Rapamycin or bedaquiline can change the response in LTR group.

RESPONSE: Indeed, for patients with resistance to drugs included in the LTR or in general patients who had failed to either a LTR or STR, a bedaquiline (Bdq) based regimen has to be considered. The current MDR-TB guidelines even recommends to replace the injectable by Bdq in the STR and we are planning to assess this strategy. However, during the period of this study, Bdq was not available. 

Rapamycin does not exist in our programme and is not also mentioned in the 2020 WHO recommendation on drug-resistant tuberculosis treatment. 

4. Immunological nitch is linked to latency / dormancy related phenotype in TB so from that analyzing few patients with foamy macrophage, MDSC , Treg along with HLA-G pattern may add to the immune mediated mechanism responsible for adverse response in LTR groups

RESPONSE: We assumed that “MDSC” stands for “myeloid-derived suppressor cells”. We are not aware of studies describing how much immunological factors explain the difference in terms of treatment response between longer and shorter treatment regimens. The aims of our study did not include this component. Moreover, the difference in terms of having a programmatically adverse outcome (either died, lost to follow-up, or treatment failure) in favor of the shorter regimen was explained by the higher proportion of patients lost to follow-up during treatment with the longer regimen. We believe that LTFU is explained by the tolerability of the treatment regimen, considering that most second-line TB drugs provoke adverse events. 

Reviewer #2: Comments:

Harouna et al. written about the treatment outcome for shorter treatment regimen and longer regimen categories with modified treatment in existing treatment regimen. Data proved the treatment utility and recommendations in present scenario for TB programme successfully progresses worldwide. Following are some suggestions which can make the manuscript stronger and beneficial for the readers:

RESPONSE: Many thanks for the feedback which improved the clarity of the manuscript

Abstract:

• Abbreviation of rifampicin resistance (RR) should come upon first appearance in the text.

RESPONSE: Ok, it has been inserted in the abstract.

Introduction:

• Introduction should modify because author had discussed previous trail related to the present study. That would be a part of discussion only. Line no. 60 to 72

RESPONSE: OK, these phrases were deleted.

• Reference No. 1 should be updated with recent one.

RESPONSE: Ok, it has been done. Now we refer to the 2019 WHO report, presenting data for 2018.

• It’s little bit inequality and confusing regarding the sentences for the dose of MFX used in the study. Line no. 57 (This study used normal-dose moxifloxacin.) indicated normal dose while, in Line no. 60 (In Guinea, the STR relied on high-dose moxifloxacin instead) high dose MFX is mentioned.

RESPONSE: Normal-dose moxifloxacin was used in the nine West and Central African countries study whereas the high-dose moxifloxacin was used in the current study.

• In first objective, author had mentioned about the diagnosis and number of RR-TB. In that view, author should write few points about the prevalence and diagnostics modalities used in his country.

RESPONSE: OK, see lines (55-57)

Method: It observed that author/s ware collected the data very precisely. However, required some information to improve manuscript like:

• Delete “longer” (line no. 99, continued the same longer regimen.).

RESPONSE: OK, it is done in the text.

• Add references regarding the doses and drugs for STR and longer regimen. (Line no. 97-106).

RESPONSE: OK

• Author should point out the type (pulmonary or extra-pulmonary) of tuberculosis among patient requited for this study.

RESPONSE: OK, see line no 96

• In patient history, alcoholism and smoker status of patient as well as house hold contact should be noted if possible. Because these habits are strongly associated with treatments which leads to recurrence.

RESPONSE: This observation is very relevant but unfortunately these data were not systematically recorded. We expanded the limitations section in the discussion as follows: “Data on conditions that could affect the outcomes, such as alcoholism, diabetes and chronic obstructive pulmonary disease, were not systematically collected, and therefore not included in the analysis.”(line 238)

Discussion:

• Provide reference for these sentences. Like:

“The STREAM trial showed 79.8% success among those treated with a….” 

RESPONSE: Ok see line 178

“We speculate that the STREAM trial setting very likely affected treatment..” We added references: RESPONSE: “Of note, the proportion LTFU in our study was very similar to the 21% reported globally (12). We speculate that the STREAM trial setting very likely affected treatment outcomes: patients who would have been LTFU in the real world were disproportionally retained in care in clinical trial conditions, in favor of the longer regimen (9).” Line 190

“However, treatment outcomes among patients treated with the shorter regimen ..” 

RESPONSE: We added a reference: “However, treatment outcomes among patients treated with the shorter regimen were less good than those reported elsewhere in the Western African region (4).” Line 204

 “But more important than the dosing of moxifloxacin, recent evidence shows that the choice of fluoroquinolone..” 

RESPONSE: Ok see line 230 “But more important than the dosing of moxifloxacin, recent evidence shows that the choice of fluoroquinolone affects treatment outcomes (16).”

• Author should modify the sentence “Another predictor of treatment success is initial resistance to fluoroquinolone”, “Bacteriologically adverse outcomes, either failure or relapse, were as frequent as 15% and 16.7%”. 

RESPONSE: Ok, we modified to “Another predictor of adverse outcomes is initial resistance to fluoroquinolone” see line 217 

• Authors are advised to cite some recent studies found relevant to the present study.

RESPONSE: We refer to a recently published review. Line 181: “Our findings are coherent with those from a recent review, which also showed that patients treated with the STR were less likely to be LTFU than those treated with the longer MDR-TB treatment regimen (reference 11).”

Conclusion: Authors are advised to reframe the sentence (Although better than outcomes from patients…..)

RESPONSE: Ok see lines 257-262: “In Guinea, programmatic success was higher when patients were treated with a high-dose moxifloxacin-based STR, when compared with a longer MDR-TB treatment regimen. However, outcomes of the STR cohort were less good than those reported elsewhere in the Western African region. Our findings, showing the reality of the National Tuberculosis Programme in Guinea, complement findings of the STREAM trial, the only other study that evaluated a high-dose moxifloxacin-based STR against a longer regimen.”

---

## [Editor Report · Decision Letter 1]

24 Jul 2020

Better programmatic outcome with the shorter regimen for the treatment of multidrug-resistant tuberculosis (MDR-TB) in Guinea: a retrospective cohort study

PONE-D-20-12232R1

Dear Dr. Hassane-Harouna,

We’re pleased to inform you that your manuscript has been judged scientifically suitable for publication and will be formally accepted for publication once it meets all outstanding technical requirements.

Kind regards,

Hasnain Seyed Ehtesham

Academic Editor

PLOS ONE

Additional Editor Comments (optional):

I have gone through this revised manuscript and also the Author response to the comments of the Reviewers. Authors have made required changes in the manuscript and added references wherever required. Authors have expanded the limitation section in the discussion part. As suggested by one of the reviewer Authors have also added few points about the prevalence and diagnostics modalities used in his country in line no.55-57. The authors have satisfactorily addressed all the comments made by the reviewers and added all required information, and have revised the manuscript accordingly. I recommend this manuscript for publication.
---

## [Editor Report · Acceptance letter]

30 Jul 2020

PONE-D-20-12232R1 

Better programmatic outcome with the shorter regimen for the treatment of multidrug-resistant tuberculosis (MDR-TB) in Guinea: a retrospective cohort study 

Dear Dr. HASSANE-HAROUNA:

I'm pleased to inform you that your manuscript has been deemed suitable for publication in PLOS ONE. Congratulations! Your manuscript is now with our production department. 

Kind regards, 

on behalf of

Prof Hasnain Seyed Ehtesham 

Academic Editor

PLOS ONE